# DAMNETS: A Deep Autoregressive Model for Generating Markovian Network Time Series

**Jase Clarkson**[*]
Department of Statistics
University of Oxford &
The Alan Turing Institute, London, UK
`jason.clarkson@stats.ox.ac.uk`

**Mihai Cucuringu**
Department of Statistics & Mathematical Institute
University of Oxford &
The Alan Turing Institute, London, UK
`mihai.cucuringu@stats.ox.ac.uk`

**Andrew Elliott**
Department of Mathematics and Statistics
University of Glasgow &
The Alan Turing Institute, London, UK
`Andrew.Elliott@glasgow.ac.uk`

**Gesine Reinert**
Department of Statistics
University of Oxford &
The Alan Turing Institute, London, UK
`reinert@stats.ox.ac.uk`

## Abstract

Generative models for network time series (also known as dynamic graphs) have tremendous potential in fields such as epidemiology, biology and economics, where complex graph-based dynamics are core objects of study. Designing flexible and scalable generative models is a very challenging task due to the high dimensionality of the data, as well as the need to represent temporal dependencies *and* marginal network structure. Here we introduce `DAMNETS`, a scalable deep generative model for network time series. `DAMNETS` outperforms competing methods on all of our measures of sample quality, over both real and synthetic data sets.

## 1 Introduction

Temporal networks (also known as dynamic graphs) arise naturally in many fields of study such as the spread of disease [1], molecular interaction networks [2], interbank liability networks [3] and online social [4] and citation networks [5]. Accurate data-driven generating modelling of these processes could have a profound wide-reaching impact, for example in simulating the trajectories of future pandemics or financial contagion risk in economic crash scenarios.

In contrast to generating static networks (i.e., networks that do not evolve over time), generating *time series of networks* has received relatively little attention in the literature. While static networks usually include complex dependencies, network time series contain complex dependencies also across time. As an example, in a time series of social contact networks, the interest may lie in replicating not only the degree distribution but also the clustering behaviour, to capture the interplay between these summary statistics over different times of the day. This complexity is further exacerbated due to the high dimensional nature of network time series; a data set with $N$ network time series on $n$ nodes each, and of length $T$ each, has size $N \times T \times n^2$. Building a generative model that faithfully replicates both network topology and dependence between graph snapshots is an extremely challenging task.

Data-driven generative models of other types of sequential data, such as natural language, commonly follow an *encoder-decoder* structure, e.g. Sequence2Sequence [6] and Transformer [7] models. We combine ideas from the static network generation and sequence modelling literatures in `DAMNETS`, an efficient and high quality generator for Markovian network time series. We leverage the insight that the *delta matrix*, that is the difference between subsequent adjacency matrices, is very sparse for

---

[*]Corresponding author

J. Clarkson et al., DAMNETS: A Deep Autoregressive Model for Generating Markovian Network Time Series. *Proceedings of the First Learning on Graphs Conference (LoG 2022)*, PMLR 198, Virtual Event, December 9–12, 2022.

most networks of interest. The key novelty of DAMNETS is that it uses a GNN to encode the current state of the network, and an efficient sparse matrix sampler to generate delta matrices conditioned on the node embeddings computed by the GNN.

In this paper, we restrict our attention to time series $G_0, G_1, \ldots, G_T$ of simple, undirected, labelled graphs on a fixed node set $V = \{1, \ldots, n\}$ with edge set $E_t \subseteq \{(i, j) : i, j \in V\}$. An element of the sequence $G_t = (V, E_t)$ has a random edge set $E_t$ drawn from a a time-dependent probability distribution $p_t(V \times V)$ over the set of node pairs on $V$, and emits adjacency matrix $A^{(t)}$.

The remainder of this paper is structured as follows. Section 2 is a review of related work. Section 3 introduces the DAMNETS algorithmic pipeline. Section 4 details the outputs of numerical experiments for representative generative models from the network literature as well as real world networks. Section 5 summarises our main findings and proposes future avenues of investigation. The DAMNETS code is available at this link.

## 2   Related Work

### 2.1   Static Network Generation

Static graph generation involves learning a probability distribution $p(G)$ over an observed set of networks. Recently, several machine learning approaches have shown good performance on generating arbitrary sets of networks, including DeepGMG [8], GraphRNN [9], GRAN [10] and BiGG [11]. Our paper continues this progression to the network time series setting.

**BiGG.**   BiGG is a scalable model for generating static networks that we will introduce briefly here, as our approach shares some similarities. Popular frameworks such as GraphRNN, GRAN and BiGG all employ the following high-level pattern for sampling the adjacency matrix; they sample each row of the adjacency matrix one at at time, using a row-wise auto-regressive model to capture the topological structure of the sampled graph and a second auto-regressive model to capture within-row edge-level correlations. GraphRNN uses a hierarchical RNN structure, GRAN uses a graph neural network with a conditional mixture of Bernoulli likelihood and BiGG uses a binary tree type structure, which is particularly suited to sparse graphs.

The major innovation introduced in BiGG is an improvement upon the naive $O(n)$ time complexity for sampling a row of the adjacency matrix. Instead of sampling each of the $n$ entries using a linear-time auto-regressive model (such as a RNN), the authors propose to sample each row using a binary tree. Each node $u$ is associated with a random binary tree $\mathcal{T}_u$ which is constructed as follows. Each tree node $k$ corresponds to an interval of graph nodes $[v_l, v_r]$. The process starts from the root $[1, n]$ and terminates at leaf nodes $[v, v]$. At each decision step the model decides whether the tree has a left child (lch), with probability $p(\text{lch}(k))$, and right child (rch), with probability $p(\text{rch}(k))$, and if so descends further down the tree until it reaches a leaf node. The probability of this tree being a particular realisation $\mathcal{T}_u = \tau_u$ is thus

$$p(\tau_u) = \prod_{k \in \tau_u} p(\text{lch}(k)) p(\text{rch}(k)). \tag{1}$$

The tree $\tau_u$ is then represented as a row vector of length $n$ of an adjacency matrix, with position $v$ having entry 1 if $\tau_u$ contains the leaf $[v, v]$, and 0 otherwise. The algorithmic advantage stems from setting all entries $[v_l, \frac{v_r}{2}]$ to 0 in row $u$ as soon as at tree node $k = [v_l, v_r]$ the left child is not generated (and similarly if a right child is not generated). Thus for a node $u$, the corresponding row of the adjacency matrix can be sampled in $O(|\mathcal{T}_u|)$ decision steps. Since $|\mathcal{N}_u|$, the size of the graph neighbourhood of $u$, equals the number of leaf nodes and $\log n$ is the maximum depth of the binary tree, the upper bound $|\mathcal{T}_u| \leq |\mathcal{N}_u| \log n$ follows. Moreover, significantly larger time savings can be made in practice if the model decides to not descend further into the tree in the upper levels.

To include dependence between entries within the row of the adjacency matrix, BiGG augments the process to produce state variables that track the decisions made, both above and below in the tree. At each tree node $k$, one always decides first whether to generate the left child conditionally on the state of the tree above, which is denoted $h_u^{top}(k)$, with the decision sampled from $p(\text{lch}(t)|h_u^{top}(k))$. If the model decides to descend into the left child, the entire left subtree is generated before returning to $t$ and making a decision about whether to generate the right child. The left subtree that was generated is summarised by a *bottom-up* state variable, denoted $h_u^{bot}(k)$, and this is used to decide whether to

sample a right child (rch) for the subtree. The model for $\mathcal{T}_u$ therefore becomes

$$p(\mathcal{T}_u) = \prod_{k \in \mathcal{T}_u} p(\text{lch}(k)|h_u^{top}(k)) \, p(\text{rch}(k)|h_u^{top}(k), h_u^{bot}(\text{lch}(k))), \quad (2)$$

where the exact equations for $h_u^{top}$ and $h_u^{bot}$ are given in Algorithm 2. The child probabilities are finally created via two MLPs, denoted $\text{MLP}_x : \mathbb{R}^F \to \mathbb{R}$ for $x = L, R$, via

$$p(\text{lch}(k) \mid h_u^{top}(k)) = \text{Bernoulli}(\text{MLP}_L(h_u^{top})(k)), \quad (3)$$

$$p(\text{rch}(k) \mid h_u^{top}(k), h_u^{bot}(\text{lch}(k))) = \text{Bernoulli}(\text{MLP}_R(h_u^{top}(k), h_u^{bot}(\text{lch}(k))). \quad (4)$$

## 2.2 Network Time Series (NTS) Generation

There are classical models for generating time series of networks designed to capture a specific set of NTS characteristics, such as the *forest fire* process [5], which can produce power-law degree distributions and shrinking effective diameter (i.e., the largest shortest path length in the graph). These classical models, while very effective at re-creating certain types of behaviour, are not data-driven and require the network to obey a pre-defined set of characteristics to be effective. Approaches attempting to generate arbitrary network time series have appeared in the machine learning literature, such as the TagGen model [12], which uses a self-attention mechanism to learn from temporal random walks on a NTS, from which new NTSs are subsequently generated. Another recent algorithm is DYMOND [13], which is a simpler approach that models the arrival times of 3-node motifs, then samples these subgraphs to generate the NTS. It is important to note that both DYMOND and TagGen attempt to solve a slightly different problem to DAMNETS; they take as input a single time series $G_0, \ldots, G_T$ and pre-defined network statistics, and aim to generate an entire time series with these network statistics similar to this single realisation. Instead of specifying the network statistics of interest, DAMNETS aims to learn a probability distribution $p(G_t|G_{t-1})$ such that given an arbitrary graph $G_{t-1}$ (not in the training set), one can draw many samples for $G_t$ and reason about the future trajectory of the network. This requires a different set of evaluation metrics and data sets, see Section 4 for discussion.

**AGE.** The approach most similar to our own is the Attention-Based Graph Evolution (AGE) model [14]. AGE uses a model very similar to a Transformer [7] (only omitting the positional encoding step), where a self-attention mechanism is applied to the rows of $A^{(t-1)}$ to learn node embeddings, and a source target attention module is sequentially applied to generate the rows of $A^{(t)}$. AGE has two clear shortcomings; the first one is that it does not explicitly account for graph connectivity, which is left to the attention mechanism to deduce. The second is that it does not capture edge-level correlations on the sampled rows. To give a simple example of why this is important, suppose we were considering a NTS where in every graph snapshot, each node has exactly two neighbours; the model should have some mechanism to condition on the edges it has sampled for a node so that it can stop once it has generated two edges. Furthermore AGE operates directly between two adjacency matrices rather than generating only differences, which does not allow it to utilise sparsity, limiting the scalability of the method. In contrast, DAMNETS explicitly utilises graph connectivity in the model pipeline and has the capacity to model edge correlations within rows of the adjacency matrix.

## 3 DAMNETS Architecture

Our goal is to learn a generative model $p(\cdot|G_{t-1})$ for the next network in a NTS, given a set of training network time series $\{\{G_t^1\}_{t=0}^{T_1}, \ldots, \{G_t^N\}_{t=0}^{T_N}, \}$. Our model has a Markovian structure and hence for generating $G_t$ all relevant information about the past is assumed to be contained in $G_{t-1}$.

For a description of our model we first introduce the *delta matrix* $\Delta^{(t)} \in \{-1, 0, 1\}^{n \times n}$ defined as

$$\Delta_{ij}^{(t)} = A^{(t)} - A^{(t-1)} = \begin{cases} 1 \implies \text{add edge } (i, j) \\ 0 \implies \text{no change in } (i, j) \\ -1 \implies \text{remove edge } (i, j). \end{cases}$$

When conditioned on $A^{(t-1)}$, each entry $\Delta_{ij}^{(t)}$ can only take *two values*, namely $\Delta_{ij}^{(t)}$ can only be 0 or 1 if $A_{ij}^{(t-1)} = 0$, and $\Delta_{ij}^{(t)}$ can only be -1 or 0 if $A_{ij}^{(t-1)} = 1$. Learning a generative model $p(\Delta^{(t)}|G_{t-1})$ is equivalent to learning $p(G_t|G_{t-1})$. Thus, this model only has to learn to produce the temporal update, rather than to reproduce the current graph *and* apply the temporal update.

As we consider only undirected graphs, we only model the lower triangular part of the delta matrix. As our approach is an encoder-decoder framework, we first summarise the previous network $G_{t-1}$

by computing node embeddings using a GNN as an encoder, then combine these with a modified version of the very efficient sparse graph sampler BiGG [11] to act as a decoder for the delta matrix.

### 3.1  The Encoder

The first step is to compute node embeddings for $G_{t-1}$, using a GNN. We employ a Graph Attention Network (GAT) [15], although any GNN layer is applicable. We use $GAT(X, A)$ to represent the application of a GAT network to a graph with node feature matrix $X$ and adjacency matrix $A$. and in the absence of other node features we use the identity matrix as node features (which here corresponds to a one-hot encoding of the nodes). Node or edge-level features, whenever available, can be incorporated into the pipeline. The embedding of $G_{t-1}$ is given by

$$H^{(t-1)} = GAT(X, A^{(t-1)}), \tag{5}$$

where $X \in \mathbb{R}^{n \times p}$ is the node feature matrix, and $H^{(t-1)} \in \mathbb{R}^{n \times q}$ is the node embedding matrix.

### 3.2  The Decoder

Starting with the first node according to the given node ordering, conditioning gives

$$p(\Delta) = p\left(\{\Delta_u\}_{u \in V}\right) = \prod_{u \in V} p\left(\Delta_u \mid \{\Delta_w : w < u\}\right).$$

We sample each row of $\Delta$ using Algorithm 2, a modified version of the BiGG row sampling algorithm. We enhance the procedure, allowing it to distinguish between a tree leaf which would be an edge addition and a tree leaf which would be an edge deletion. If the left (resp. right) child at level $k$ is a leaf node corresponding to entry $\Delta_{ij}^{(t)}$, instead of (3) we sample the leaf node using

$$p(\mathrm{lch}(k) \mid h) = \begin{cases} \mathrm{Bernoulli}(\mathrm{MLP}_+(h)) \text{ if } A_{ij}^{(t)} = 0, \\ \mathrm{Bernoulli}(\mathrm{MLP}_-(h)) \text{ if } A_{ij}^{(t)} = 1, \end{cases} \tag{6}$$

where $h \in \mathbb{R}^q$ is the corresponding state variable. Each application of Algorithm 2 returns an embedding, namely $g_u = h_u^{bot}(root)$ which depends on every entry in the row. As is done in the static setting we apply an auto-regressive model across these row embeddings to capture dependencies between rows. The bottom-up embeddings of each tree have no other computational dependencies, so can be efficiently pre-computed during training. We chose to use a standard Transformer self-attention layer [7] (which we call TFEncoder) with sinusiodal positional embedding for this auto-regressive component; this was chosen to provide similar representation power to the baseline model AGE. Self attention does not scale to very long sequences however, so for very large graphs with many nodes, this could be replaced by either an LSTM or the *Fenwick Tree* structure proposed in [11].

### 3.3  The DAMNETS model architecture

**Figure 1:** An overview of our approach to generating Markovian transitions in a network time series. We learn a generative model of the lower triangular part of the *delta matrix* given the previous graph $G_{t-1}$. We then draw a sample $\Delta^{(t)}$ and add this to $A^{(t-1)}$ to produce a sample $G_t$.

With the two key components of our model defined, we now explain how these models are combined to generate delta matrices given an input graph. As stated in Equation (5), we first compute node embeddings $H^{(t-1)} \in \mathbb{R}^{n \times F}$, with $H_i^{(t-1)} \in \mathbb{R}^F$ representing the node embedding computed for node $i$ in $G_{t-1}$. When generating the row tree $\mathcal{T}_u$ for node $u$, (which corresponds to generating the row of the delta matrix for node $u$), we combine the node embedding from the previous network with the row-wise auto-regressive term $h_{u-1}^{row}$ computed by TFEncoder via an MLP

$$h_u^{top}(root) = \mathrm{MLP}_{cat}(h_{u-1}^{row}, H_u^{(t-1)}). \tag{7}$$

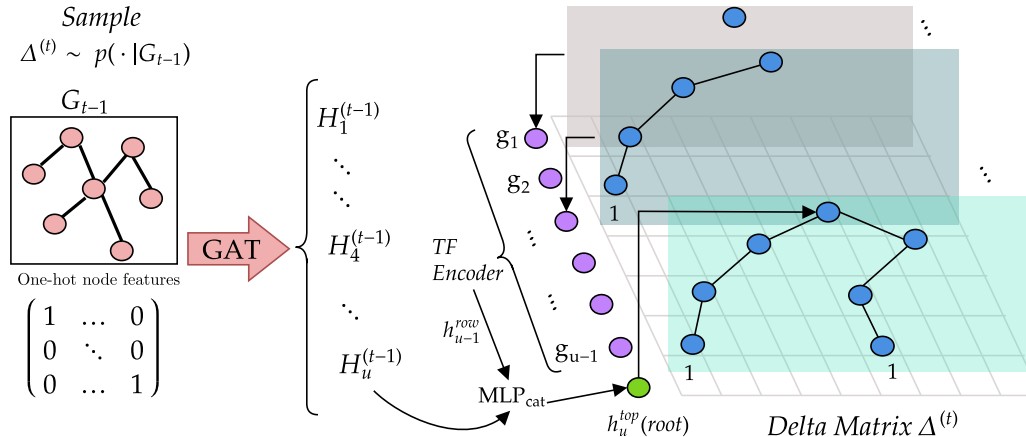

**Figure 2:** A visualisation of the generation of the $u$-th row of the delta matrix $\Delta^{(t)}$ using the `DAMNETS` model architecture. The nodes shown in red indicate the graph $G_t$. We use a GAT to compute node embeddings $H^{(t)}$ for each node in $G_t$. Nodes shown in blue belong to the binary tree generated for each row; each tree is generated by combining the node embedding in the previous graph with an auto-regressive term computed using a Transformer (TF) Encoder across the rows of the delta matrix to produce $h_{u-1}^{row}$, which is used in Equation (7) to initialise the top-down descent of each tree.

where $\mathrm{MLP}_{cat} : \mathbb{R}^{2F} \to \mathbb{R}^F$. The full procedure is described in Algorithm 1, with a detailed version Algorithm 2 in the SI, and is visualised in Figures 1 and 2. The model is trained via maximum likelihood over the entries of the delta matrix using gradient descent. The advantage of this framework is twofold; firstly the delta matrix is usually much sparser than the full adjacency matrix, allowing us to well utilise sparse sampling methods. This is a very natural assumption: one does not expect *most* of the network to change at each timestep, but rather just a small subset of the edges. The second is that differencing a time series makes learning easier. It is very common in traditional time series analysis to perform differencing transformations on data, as differencing may alleviate trends in the time series.

---

**Algorithm 1:** Algorithm for generating the the delta matrix $\Delta^{(t)}$ using `DAMNETS`

---

**Input:** Input graph $G_{t-1} = (V, E_{t-1})$, node features $X$
$H^{(t-1)} \leftarrow GAT(X, A^{(t-1)})$
$h_0^{row} \leftarrow \emptyset$
**for** $u \leftarrow 1$ **to** $n$ **do**
  Let $k = \{1, \ldots, u-1\}$ be the root of tree $\mathcal{T}_u$.
  $h_u^{top}(k) = \mathrm{MLP}_{cat}(h_{u-1}^{row}, H_u^{(t-1)})$.
  $g_u, \mathcal{N}_u \leftarrow \texttt{Recursive}(u, k, h_u^{top}(k))$             `/* Algorithm 2 */`
  `/* Only non-zero indices are returned in` $\mathcal{N}_u$ `*/`
  $\Delta_u \leftarrow$ Determine sign of entries using $A^{(t-1)}$ and transform into a vector.
  $h_u^{row} \leftarrow \mathrm{TFEncoder}(g_u; g_{1:u-1})$
**end**
**Return** $\Delta^{(t)}$ with rows $\Delta_u$, $u = 1, \ldots, n$.

---

## 4 Experiments

Evaluating a generative model usually follows the following recipe: fit the generative model on the training data, draw samples from the model and then compare the distribution of these samples to some held out test data using some kind of statistical test or metric on the space of probability distributions. For static graphs, there exist a number of *graph kernels* [16] from which a Maximum Mean Discrepancy (MMD) [17] type metric can be derived. However these are very computational costly (some scaling as $O(n^4)$ for a graph with $n$ nodes). It is therefore common to define a set of

*summary statistics* over the graphs, such as the degree distribution or clustering coefficient distribution, and compare the distributions of these summary statistics computed over the sampled and test graphs.

We adopt a similar approach applied to the marginal distributions of the network time series. We choose to compare six different network statistics, three local and three global (see [18] for a background on network statistics). Our three local properties are the degree distribution, clustering coefficient distribution and the eigenvalue distribution of the graph Laplacian as introduced in [10]. For each graph, we compute a histogram of these properties over the nodes in the graph, and use a Gaussian kernel with the total-variation metric to compute the MMD. Our three global measures are transitivity, assortativity and closeness centrality. Each of these metrics produces one scalar value per graph, and we again use a Gaussian kernel with the $\ell^2$ metric to compute the MMD.

For each time point $t$ and statistic $S(\cdot)$, we compute $\mathrm{MMD}_t(S(G_t^{test}), S(G_t^{sampled}))$, and use as final metric the sum $\overline{\mathrm{MMD}}(S) = \sum_t \mathrm{MMD}_t(S(G_t^{test}), S(G_t^{sampled}))$. If the marginal distributions match exactly, $\overline{\mathrm{MMD}}(S)$ will equal 0, and smaller values indicate better agreement between the distributions. We display all $\overline{\mathrm{MMD}}$ scores to three significant figures. Comparing the marginal distributions alone does not suffice as a comparison metric, so we also provide summary plots of these network statistics through time to verify that the evolution of these statistics match. In addition, we have designed several synthetic-data experiments to verify specific time-series properties observed in real-world networks which we would like to capture.

A difficulty for graph generative model evaluation is that proper comparison of a network time series generator requires many realisations of this time series drawn from the same distribution to facilitate learning and subsequent comparison. Papers such as TagGen [12] and DYMOND [13] utilise data sets that comprise of one realisation of a real world temporal network, and aim to simply produce *"surrogate"* networks that closely resemble that single realisation. We aim to assess whether our model is able to generalise to new examples, in the sense that given a new graph $G_{t-1}$ drawn from the same distribution as the training distribution, we can draw samples from $G_t \sim p(\cdot|G_{t-1})$. We are therefore unable to use the same data sets as these papers, and instead design a new experimental setup in line with our objective.

Our general experimental framework is as follows: we are given a set of realisations $\{\{G_t^1\}_{t=0}^{T_1}, \ldots, \{G_t^N\}_{t=0}^{T_N}, \}$. For DAMNETS and AGE, we split this up into a set of training time series and test time series, and fit each model on the training set, then evaluate the performance on the test set. As DYMOND and TagGen can only learn from one time series at a time and produce realisations from that specific time series, we instead train an instance of these models separately on each time series in the test set and sample one time series from each trained model. This might seem like a large advantage for these models, as they have direct access to the test set. However our experimental results show that the aggregated behaviour of these samples does not match the underlying distribution well, suggesting these methods are not suitable for learning the true underlying process that a given sample was drawn from. Due to the fact that DYMOND and TagGen have to be re-trained on every single time series, we provide two sets of results for some data sets, with a smaller data set chosen such that DYMOND and TagGen converge within 24 hours.

## 4.1 The Barabási–Albert Model

The family of Barabási–Albert (B-A) models [19] was designed to capture the so-called *scale-free* property observed in many real world networks through a preferential attachment mechanism. Formally a scale-free network is one whose degree distribution follows a power-law; if $\deg(i)$ represents the degree of node $i$ in a random network model, then the network is scale free if $\mathbb{P}(\deg(i) = d) \propto \frac{1}{d^\gamma}$, for some constant $\gamma \in \mathbb{R}$. Degree distributions with a power-law tail have been observed in many real networks of interest, such as hyperlinks on the World-Wide Web or metabolic networks, although the ubiquity of power law degree distributions has been disputed [20].

The B-A model has two integer parameters, the number of nodes $n$ and the number of edges $m$ to be added at each iteration. The network is initialised with $m$ initial connected nodes. At each iteration $t$, a new node is added and is connected to $m$ existing nodes, with probability proportional to the current degree $p_u = \frac{\deg(u)}{\sum_{v \in V} \deg(v)}$. Here, the standard NetworkX [21] implementation is used. Constructing a B-A network in this way yields a network time series of length $T = n - m$, where each graph $G_t$ is the graph after node $m + t$ has the first edges attached to it. Nodes with a many existing connections

(known as *hubs*) will likely accumulate more links; this is the *preferential attachment* property which, in the B-A model, leads to a power-law degree distribution with scale parameter $\gamma = 3$.

For the B-A experiments, we take $N = 200$ time series with parameters $n = 100$ and $m = 4$, yielding time series of length $T = 96$. The results are displayed in Table 1 and Figure 3. We see DAMNETS produces samples with orders of magnitude lower $\overline{\text{MMD}}$ than the baseline methods, and is the only model to correctly replicate the power law degree distribution.

**Table 1:** The $\overline{\text{MMD}}$ on the B-A dataset for each network statistic. Lower is better.

| Model | Degree | Clustering | Spectral | Transitivity | Assortativity | Closeness |
|---|---|---|---|---|---|---|
| DYMOND | 14.01 | 61.20 | 8.78 | 7.28 | 4.76 | 3.19 |
| TagGen | 16.33 | 16.55 | 2.29 | 2.06 | 23.95 | 0.10 |
| AGE | 15.08 | 25.15 | 9.45 | 3.42 | 6.37 | 2.36 |
| DAMNETS | $\mathbf{8e^{-3}}$ | **0.78** | **0.14** | **0.01** | **0.01** | $\mathbf{5e^{-6}}$ |

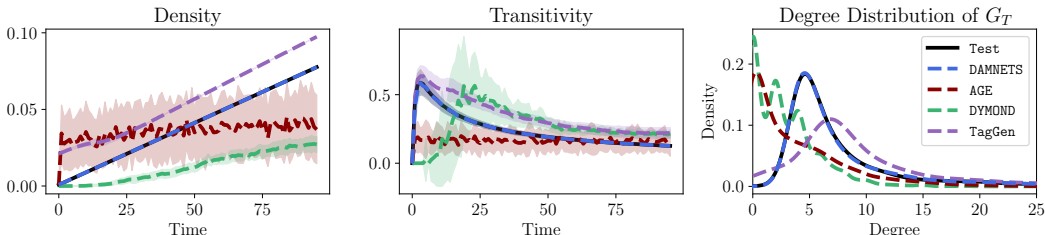

**Figure 3:** Plots for the B-A model. Left: density against time; middle: transitivity against time; right: the average degree distribution of the final network $G_T$ produced by the models. Only DAMNETS correctly replicates the power law degree distribution.

## 4.2 Bipartite Concentration

**Figure 4:** A sample from the bipartite concentration model with 10 nodes in each partition, with an initial connection probability of $p = 0.2$ and a concentration proportion $p^{con} = 0.3$. The highest degree node is shown in red; links concentrate on this node over time.

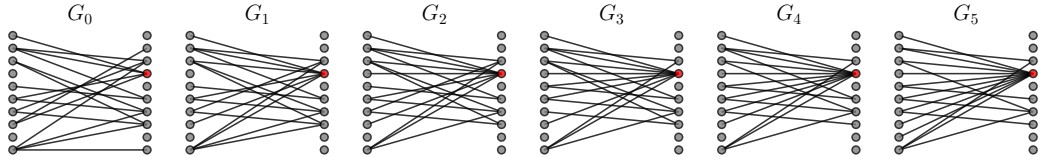

This data set is designed to simulate behaviour in rating systems where objects with many links tend to accumulate more recommendations [22]. For example in a data set consisting of users and movies, movies with many existing recommendations are likely to accumulate more over time. The graph $G_0$ is initialised as a random bipartite graph with connection probability $p$. At each timestep, we select the node in the right-hand partition with the most links (ties broken at random) and re-wire a proportion $p^{con}$ of non-adjacent edges to that node.

For the experiments we set $p = 0.5$ and $p^{con} = 0.1$. For the smaller data set (S), we place 30 nodes in each partition (so $n = 60$) and iterate for $T = 10$ timesteps. For the larger data set (L) we place 250 nodes in each partition ($n = 500$) and iterate for $T = 15$ timesteps. To measure the extent to which the different generators replicate this bipartite structure, in addition to our standard summaries we also compute the mean Spectral Bipartivity (SB) [23] through time, which takes values in $[0, 1]$, with 0 indicating the network is not bipartite and 1 indicating the network is fully bipartite. The results are displayed in Table 2 and Figure 9. DAMNETS consistently outperforms all the baseline models across all summary statistics.

**Table 2:** The $\overline{\mathrm{MMD}}$ for each network statistic (lower is better) and Spectral Bipartivity (closer to 1 is better) across the small (S) and large (L) bipartite contraction test datasets.

| Model | Deg. | | Clust. | | Spec. | | Trans. | | Assort. | | Closeness | | SB | |
|---|---|---|---|---|---|---|---|---|---|---|---|---|---|---|
| | (S) | (L) | (S) | (L) | (S) | (L) | (S) | (L) | (S) | (L) | (S) | (L) | (S) | (L) |
| DYMOND | 1.06 | – | 9.55 | – | 0.12 | – | 1.67 | – | $9e^{-4}$ | – | 0.14 | – | 0.50 | – |
| TagGen | 0.81 | – | 1.73 | – | 0.29 | – | $5e^{-4}$ | – | 0.07 | – | $2e^{-4}$ | – | 0.56 | – |
| AGE | 0.92 | 2.75 | 9.46 | 15.3 | 0.13 | 0.25 | 1.48 | 3.71 | 0.72 | 4.81 | 0.16 | 0.36 | 0.55 | 0.52 |
| DAMNETS | **0.01** | **$4e^{-3}$** | **0.11** | **$3e^{-3}$** | **0.03** | **$5e^{-4}$** | **$7e^{-6}$** | **$8e^{-8}$** | **$1e^{-4}$** | **$7e^{-6}$** | **$4e^{-7}$** | **$1e^{-7}$** | **0.99** | **0.99** |

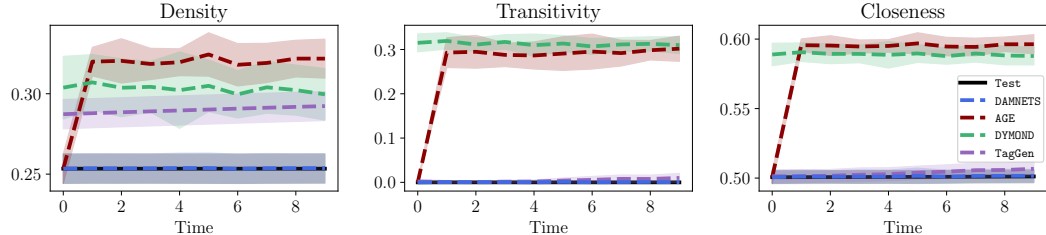

**Figure 5:** Plots for the bipartite contraction model. Left: density against time; middle: transitivity against time; right: closeness against time. Only DAMNETS shows good performance in all statistics.

## 4.3 Community Evolution and Decay

**Figure 6:** A sample from the community decay model of length $T = 5$ on $V = \{1, \ldots, 45\}$, with 15 nodes in each of the $Q = 3$ communities, connection probabilities $p_{int} = 0.7$, $p_{ext} = 0.005$, decay community $D = 3$ (coloured red) and decay proportion $p_{dec} = 0.2$.

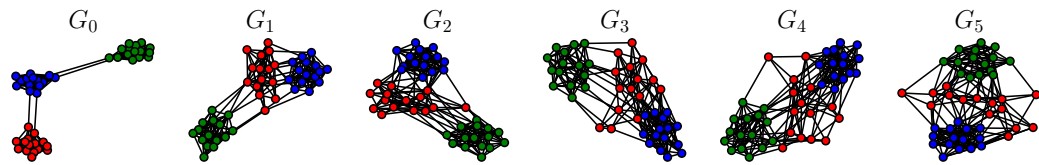

Our next network time series benchmark considers a dynamic community structure model. We initialise a three-community stochastic block model on $n$ nodes. At each time step, we re-wire a fixed proportion $f_{dec}$ of the third community (which we call the decay community), replacing them with a random outgoing edge to a node in one of the other communities. A sample from the model is shown in Figure 6, and a full description of the model is given in Appendix A.2.

For the experiments we use inter-community connection probability $p_{int} = 0.9$, intra-community $p_{ext} = 0.01$, decay fraction $f_{dec} = 0.2$ and iterate for $T = 20$ timesteps. For the small (S) data set we place 20 nodes in each community (for a total of $n = 60$ nodes) and for the large (L) data set we place 400 nodes in each community ($n = 1200$ in total). The non-decay communities should have constant density, and the decay community should have density decaying exponentially at rate $f_{dec}$. The results are displayed in Table 3 and Figure 10. DAMNETS is the best performing model overall, although AGE also shows strong performance on this data set.

**Table 3:** The $\overline{\mathrm{MMD}}$ for each network statistic across the small (S) and large (L) community decay test data sets, with a $(-)$ when the model did not converge within 24 hours. A lower $\overline{\mathrm{MMD}}$ is better.

| Model | Deg. | | Clust. | | Spec. | | Trans. | | Assort. | | Closeness | |
|---|---|---|---|---|---|---|---|---|---|---|---|---|
| | (S) | (L) | (S) | (L) | (S) | (L) | (S) | (L) | (S) | (L) | (S) | (L) |
| DYMOND | 1.95 | – | 3.20 | – | 0.66 | – | 0.88 | – | 1.02 | – | 0.33 | – |
| TagGen | 10.99 | – | 2.91 | – | 2.18 | – | 0.26 | – | 2.37 | – | 1.04 | – |
| AGE | **0.15** | **0.17** | 2.00 | 2.06 | 0.43 | 0.42 | 0.02 | 0.03 | 0.07 | 0.06 | **0.01** | 0.03 |
| DAMNETS | 0.19 | 0.21 | **1.90** | **1.91** | **0.39** | **0.40** | **0.01** | **0.01** | **0.03** | **0.04** | **0.01** | **0.02** |

**Figure 7:** The density of each community through time in the 3-community dataset.

## 4.4 Correlation Networks

This data set consists of financial correlation networks built from time series of asset prices from the Wharton CRSP database [24]. We consider a set of 49 liquid stocks from the US equity market, for which we have available minutely prices data. We construct a graph by assigning each stock to a node. We then estimate the correlation matrix of their 5-minute returns each day, and threshold these correlations at 1 standard deviation in order to construct the edges (so stocks are connected by an edge if they are strongly correlated). The data set spans $N = 97$ weeks, with each week giving a time series of length $T = 5$.

One issue with this data set is that correlations between financial instruments are known to be unstable over time (hence different realisations may not drawn from the same distribution). To mitigate this we did not split the data chronologically, but have rather drawn the training and test splits randomly (which correspond to selecting random weekly time series from the data set). We repeat this procedure over 5 seeds and compute the average $\overline{\mathrm{MMD}}$. The results are displayed in Table 4 and Figure 11. DAMNETS is the only model to show good performance across all statistics.

**Table 4:** The $\overline{\mathrm{MMD}}$ for each network statistic across the correlation test data set. Lower is better.

| Model | Degree | Clustering | Spectral | Transitivity | Assortativity | Closeness |
|---|---|---|---|---|---|---|
| DYMOND | 0.16 | 0.58 | 0.27 | 0.17 | 0.04 | 0.06 |
| TagGen | 0.95 | 0.56 | 0.85 | $4e^{-3}$ | 0.08 | 0.48 |
| AGE | 0.14 | 1.07 | 0.31 | 0.26 | 0.08 | 0.10 |
| DAMNETS | **0.13** | **0.21** | **0.25** | 0.04 | **0.02** | **0.01** |

## 4.5 The MIT Reality Mining Dataset

This is a contact network between students and faculty at the MIT media lab recorded between August 2004 to May 2005 [25]. Each contact between two different people (recorded via a bluetooth device on the subjects' mobile phones) forms a timestamped edge. We aggregated all daily contacts into networks, and evaluate our procedure on generating weekly time series of these contact networks. We dropped weeks without observations for all the days, giving a set of 32 weekly time series. We used 16 weeks for training, 6 for validation and 10 for testing. As with the correlation data set, we randomly sample weeks to form the train and test set, and repeat the experiment across three seeds. The results in Table 5 show that DAMNETS performs best on all statistics except closeness, even compared to DYMOND and TagGen which have access to the test data at training. The strong performance of DAMNETS is particularly evident across the local summary statistics, which suggests DAMNETS is particularly well suited to represent fine-grained local structure.

**Table 5:** The $\overline{\mathrm{MMD}}$ for each network statistic across the MIT test data set. Lower is better.

| Model | Degree | Clustering | Spectral | Transitivity | Assortativity | Closeness |
|---|---|---|---|---|---|---|
| DYMOND | 1.24 | 2.21 | 1.28 | 0.39 | 0.18 | **0.04** |
| TagGen | 2.57 | 2.99 | 2.42 | 0.54 | 0.32 | 0.48 |
| AGE | 2.02 | 2.75 | 2.17 | 0.37 | 0.38 | 0.73 |
| DAMNETS | **0.41** | **1.42** | **0.46** | **0.34** | **0.10** | 0.09 |

## 4.6 Ablation Study

We see that DAMNETS outperforms all the baseline models on each data set under consideration, in particular the AGE model, which is the most similar in that it also follows a Sequence2Sequence framework. DAMNETS differs from AGE in two major ways, namely the formulation in terms of the

delta matrix and the model architecture adapted for sampling this sparse matrix. We provide an ablation study in Appendix B where we modify AGE to generate delta matrices, and also a version where we add positional encodings. We find that the delta matrix formulation significantly improves the performance of AGE, while positional encodings do not change the performance much, with neither variant of AGE able to match the performance of DAMNETS. This suggests it is the combination of our re-formulation of the problem combined with a model architecture suited to sample sparse delta matrices that provides such strong performance. We also provide separate experiments to examine the influence of the GNN layer type, GNN depth and type of recurrent module in the decoder.

## 5 Discussion and Conclusion

DAMNETS provides a novel approach to generating network time series, with the ability to have fine-grained edge-level conditioning while maintaining scalability by generating delta matrices rather than entire graphs and efficiently utilising the sparsity of these matrices. We have shown through extensive experiments that DAMNETS is able to learn a variety of important network models that existing methods simply cannot. DAMNETS can learn to generate long time series, reproduce power-law degree distributions, bipartite structure and maintains very strong performance on larger networks, while none of the baseline models are able to capture all of these properties.

In future work, the Markovian assumption underlying DAMNETS could be relaxed to incorporate time series with long range dependencies, using techniques such as *node memory* introduced in the TGNN model [26]. The model could also be extended to handle graphs of varying size: node deletion could be performed by adding a step before the sampling of each row-tree wherein the model makes a decision about whether the node should persist to the current timestep. Node additions could be handled by allowing optional rows to be appended at the end of the delta matrix (and only sampling ones for these rows, as a new node could not have any edge deletions). It would also be interesting and fairly straightforward to extend DAMNETS to generate node attributes, along the lines of [27].

## Acknowledgements

J.C. acknowledges funding from the EPSRC CDT in Modern Statistics and Statistical Machine Learning (EP/S023151/1), and The Alan Turing Institute's Finance and Economics Programme. M.C. acknowledges support from the EPSRC grants EP/N510129/1 and EP/W037211/1 at The Alan Turing Institute. A.E. is supported by The Alan Turing Institute's Finance and Economics Programme and in part by EPSRC grant EP/W037211/1 at The Alan Turing Institute. G.R. is supported in part by EPSRC grants EP/T018445/1, EP/W037211/1 and EP/R018472/1.

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

## A  Supplementary Information for DAMNETS: A Deep Autoregressive Model for Generating Markovian Network Time Series

### A.1  The `DAMNETS` Row Generation Algorithm

---

**Algorithm 2:** Algorithm for generating the the $u^{th}$ row of the delta matrix

---

**Function** `Sample_Leaf`($u$, $k$, $h$)**:**

   $e \leftarrow (u, k)$

   **if** $e$ is Edge Addition **then**

      has_leaf $\sim$ Bernoulli(MLP$_+(h)$)

   **else**

      has_leaf $\sim$ Bernoulli(MLP$_-(h)$)                           /* Edge deletion */

   **end**

   **if** has_leaf **then**

      **return** $\vec{1}, e$

   **else**

      **return** $\vec{0}, \emptyset$

   **end**

**End Function**

**Function** `Recursive`($u$, $k$, $h_u^{top}(k)$)**:**

   **if** is_leaf($\text{lch}_u(k)$) **then**

      $h_u^{bot}\left(\text{lch}_u(k)\right), \mathcal{N}_u^{k,left} \leftarrow$ `Sample_Leaf`$(u, \text{lch}_u(k), h_u^{top}(k))$

   **else**

      *has_left* $\sim$ Bernoulli(MLP$_L(h_u^{top}(k))$)

      **if** has_left **then**

         $h_u^{top}(\text{lch}_u(k)) \leftarrow$ LSTMCell$\left(h_u^{top}(k), \text{embed(left)}\right)$

         $h_u^{bot}\left(\text{lch}_u(t)\right), \mathcal{N}_u^{k,left} \leftarrow$ `Recursive`$(u, \text{lch}_u(k), h_u^{top}(\text{lch}_u(k)))$

      **else**

         $h_u^{bot}\left(\text{lch}_u(k)\right), \mathcal{N}_u^{k,left} \leftarrow \vec{0}, \emptyset$

      **end**

   **end**

   $\hat{h}_u^{top}(\text{rch}(k)) \leftarrow$ TreeCell$^{top}\left(h_u^{bot}(\text{lch}_u(k)), h_u^{top}(\text{lch}_u(k))\right)$

   **if** is_leaf($\text{rch}_u(k)$) **then**

      $h_u^{bot}\left(\text{rch}_u(k)\right), \mathcal{N}_u^{k,right} \leftarrow$ `Sample_Leaf`$\left(u, \text{rch}_u(k), \hat{h}_u^{top}(\text{rch}_u(k))\right)$

   **else**

      *has_right* $\sim$ Bernoulli(MLP$_L(\hat{h}_u^{top}(\text{rch}_u(k)))$)

      **if** has_right **then**

         $h_u^{top}(\text{rch}(k)) \leftarrow$ LSTMCell$\left(\hat{h}_u^{top}(\text{rch}(k)), \text{embed(right)}\right)$

         $h_u^{bot}\left(\text{rch}_u(k)\right), \mathcal{N}_u^{k,right} \leftarrow$ `Recursive`$(u, \text{rch}_u(k), h_u^{top}(\text{rch}_u(k)))$

      **else**

         $h_u^{bot}\left(\text{rch}_u(k)\right), \mathcal{N}_u^{k,right} \leftarrow \vec{0}, \emptyset$

      **end**

   **end**

   $h_u^{bot}(k) \leftarrow$ TreeCell$^{bot}\left(h_u^{bot}\left(\text{lch}_u(k)\right), h_u^{bot}\left(\text{rch}_u(k)\right)\right)$

   $\mathcal{N}_u^k \leftarrow \mathcal{N}_u^{k,left} \cup \mathcal{N}_u^{k,right}$

   **return** $h_u^{bot}(k), \mathcal{N}_u^k$

   **End Function**

---

First we provide details for the `DAMNETS` row generation algorithm given in Algorithm 2. Here, TreeCell$^{bot}$ and TreeCell$^{top}$ are two TreeLSTM [28] cells, embed(left) and embed(right) are learned embeddings for the binary values "left" and "right", and LSTMCell is a standard LSTM [29]. The top down cell summarises decisions made above $t$ in the tree, and the bottom up cell summarises lower levels of the tree (if they exist), where $h_u^{bot}(\emptyset) = 0$. Notice that that $h^{bot}$ is computed independently of $h^{top}$.

## A.2  The Community Decay Model

The three community decay model is formally defined as follows. The initial network $G_0 = (V, E_0)$ with node set $V = \{1, \ldots, n\}$ is equipped with a surjective community membership function $C : \{1, \ldots, n\} \rightarrow \{1, \ldots, Q\}$ that encodes which of the $Q$ communities a given node $i$ belongs to (a node can only belong to one community). Here we assume that the community memberships are known. The initial graph $G_0$ is then fully described by the *interior* (within community) and *exterior* (across communities) edge probabilities $p_{ij} := \mathbb{P}((i, j) \in E_0)$, given by

$$p_{ij} = \begin{cases} p_{int} & \text{if } C(i) = C(j) \\ p_{ext} & \text{if } C(i) \neq C(j). \end{cases} \tag{8}$$

A network time series $G_1, \ldots, G_T$ is then constructed as follows; we fix a community $D \in \{1, \ldots, Q\}$ as the *decay* community. We define the set of *internal* edges for community $D$ as

$$D_t^{int} := \{(i, j) \in E_t \mid C(i) = C(j) = D\}. \tag{9}$$

At each iteration $t$, (i.e time step), a fixed proportion $f_{dec}$ of the *internal* edges $D_t^{int}$ are replaced with *external* edges. This is achieved by selecting a random internal edge $(i, j)$ and removing it from the edge set $E_t$, then selecting a node $u$ uniformly from $\{i, j\}$. We then select a random endpoint $k$ uniformly from $\{v \in V \mid C(v) \neq D, (u, v) \notin E_t\}$, the set of nodes not in community $D$ and not connected to $u$, and finally add the edge $(u, k)$ to the edge set $E_t$. We repeat this procedure $T$ times to generate our network time series.

The model can be interpreted as starting with a network with $Q$ densely connected communities, decaying in time to have only $Q - 1$ clear communities; the decay community $D$ will appear as noise around those left unperturbed. A sample from the model can be seen in Figure 6; for ease of visualisation, each initial community has only 15 nodes.

## A.3  Graph Attention Networks

For the encoder step in DAMNETS we compute node embeddings for $G_{t-1}$, using a GNN. We employ a Graph Attention Network (GAT) [15], although any GNN layer is applicable. Given node features $X_1, \ldots, X_n, X_i \in \mathbb{R}^F$, a GAT layer produces a new set of node features $h_i \in \mathbb{R}^{F'}$ according to

$$h_i = \sigma \left( \sum_{j \in \mathcal{N}_i} \alpha_{ij} W X_j \right), \tag{10}$$

where $W \in \mathbb{R}^{F' \times F}$ is a learnable weight matrix, $\sigma(\cdot)$ is a non-linear function applied element-wise, and $\alpha_{ij} \in \mathbb{R}$ are normalised attention coefficients computed as

$$e_{ij} = a(W X_i \| W X_j), \tag{11}$$

$$\alpha_{ij} = \frac{\exp(e_{ij})}{\sum_{k \in \mathcal{N}_i} \exp(e_{ik})}, \tag{12}$$

where $\|$ represents the concatenation operation, and $a(\cdot)$ is a single layer MLP with the LeakyReLU activation function. These layers are stacked to produce a GAT network. GAT layers can also employ *multi-head* attention [7]. We write $GAT(X, A)$ to represent the application of a GAT network to a graph with node feature matrix $X$ and adjacency matrix $A$.

## A.4  Further Related Work

**Network Time Series Forecasting.**  A distinct but related area of study is network time series forecasting, where the goal is to predict node attributes or links in a graph at a future time point. Classical approaches include the GNAR model [30] which assumes a simple linear model based on lagged network attributes. Many approaches have appeared in the deep learning literature, such as the Graph AR model [31] and Variational Graph Recurrent Neural networks [27]. Markov models have been considered in this literature before, in particular the Graph Edit Network model [32], which uses a GNN encoder to predict a list of insertions and deletions for both nodes and edges at the next timestep, which can be viewed as an estimate of the delta matrix at the next time step. It remains to note that forecasting focuses on the next time (points) and does not produce a whole time series of networks which resembles the observed time series.

# B Ablation Studies

## B.1 The Delta Parameterisation

In many of our experiments, `DAMNETS` outperforms the `AGE` model [14]. One may conjecture that it is the use of the delta matrix which is the main driver of this difference in performance. To assess this hypothesis we perform the following ablation study: we re-formulate the `AGE` model to generate delta matrices instead of entire adjacency matrices. Recall that `AGE` is a transformer model as described in [7], but without the positional encodings. The transformer is trained via maximum likelihood to generate rows of the adjacency matrix $A^{(t+1)}$ from the rows of the previous adjacency $A^{(t)}$ using the standard Sequence2Sequence framework.

We instead re-formulate `AGE` to generate delta matrices, which we call `AGE-D`. To ensure valid delta matrices, we propose to train a transformer to generate $|\Delta^{(t)}|$ row-wise from the rows of $A^{(t)}$, then construct $A^{(t+1)}$ as

$$A^{(t+1)} = \left\{ A^{(t)} + \left| \Delta^{(t)} \right| \right\} \mod 2.$$

Note that this always produces a valid adjacency matrix. We also include a variant with positional encodings on both the input and output rows, which we title `AGE-DPE`. We compare the performance of `AGE` and the two proposed variants on the BA and the Bipartite Contraction (L) datasets, as there was a particularly large gap in performance between `DAMNETS` and `AGE` on these datasets. We also include the $\overline{\text{MMD}}$ for `DAMNETS` again for ease of reference. The results are displayed in Tables 6 and 7.

**Table 6:** The $\overline{\text{MMD}}$ for each network statistic on the BA dataset. Lower is better.

| Model | Degree | Clustering | Spectral | Transitivity | Assortativity | Closeness |
|---|---|---|---|---|---|---|
| AGE | 15.08 | 25.15 | 9.45 | 3.42 | 6.37 | 2.36 |
| AGE-D | 0.76 | 2.45 | 0.69 | 0.51 | 4.52 | $2e^{-3}$ |
| AGE-DPE | 0.76 | 2.37 | 0.71 | 0.49 | 4.31 | $2e^{-3}$ |
| DAMNETS | $\mathbf{8e^{-3}}$ | **0.78** | **0.14** | **0.01** | **0.01** | $\mathbf{5e^{-6}}$ |

**Table 7:** The first block shows the $\overline{\text{MMD}}$ for each network statistic on the Bipartite Contraction (L) dataset, for which lower is better. The last column shows the spectral bipartivity, for which a value closer to 1 is better.

| Model | Degree | Clustering | Spectral | Transitivity | Assortativity | Closeness | SB |
|---|---|---|---|---|---|---|---|
| AGE | 2.75 | 15.3 | 0.25 | 3.71 | 4.81 | 0.36 | 0.52 |
| AGE-D | 0.15 | 6.14 | 0.15 | 0.04 | 0.02 | $1e^{-2}$ | 0.85 |
| AGE-DPE | 0.13 | 6.07 | 0.17 | 0.03 | 0.02 | $1e^{-2}$ | 0.87 |
| DAMNETS | $\mathbf{4e^{-3}}$ | $\mathbf{3e^{-3}}$ | $\mathbf{5e^{-4}}$ | $\mathbf{8e^{-8}}$ | $\mathbf{7e^{-6}}$ | $\mathbf{1e^{-7}}$ | **0.99** |

We see that re-formulating `AGE` to generate delta matrices significantly improves the performance of the method on these datasets, whilst adding the positional encodings provided little to no gain in performance. We also note that while `AGE-D` performs better than `AGE`, it still does not match the performance of `DAMNETS` on these datasets, indicating that it is the combination of our formulation of the problem *and* specific choice of architecture that leads to such strong performance.

## B.2 GNN Layer Type

Here we study the impact on the performance of `DAMNETS` when using different types of GNN layers in the encoder. The three layers we consider are GAT [15] used in the main text, GCN [33] and GraphSAGE [34] with mean aggregation. We repeat the BA experiment from the main text using a single GNN layer of each type, and report the $\overline{\text{MMD}}$ statistic for each variation in Table 8. We see that `DAMNETS` is not particularly sensitive to the choice of GNN layer.

**Table 8:** The $\overline{\text{MMD}}$ for each network statistic on the BA dataset for samples generated by `DAMNETS` using different GNN encoder layers. In each case we use a single layer GNN. Lower is better. We see that `DAMNETS` is not particularly sensitive to the choice of GNN.

| Model | Degree | Clustering | Spectral | Transitivity | Assortativity | Closeness |
|---|---|---|---|---|---|---|
| GAT | $8e^{-3}$ | **0.78** | **0.14** | **0.01** | **0.01** | **$5e^{-6}$** |
| GCN | $8e^{-3}$ | 0.81 | 0.17 | 0.02 | **0.01** | $7e^{-6}$ |
| GraphSAGE | **$7e^{-3}$** | 0.80 | 0.16 | **0.01** | 0.02 | $7e^{-6}$ |

### B.3 GNN Depth

We repeat the BA experiment from the main text varying the depth of the GNN. We use GAT [15] layers for the encoder with a depth of 1, 2 and 4 layers respectively. The results are displayed in Table 9. We see that performance slightly degrades with deeper networks, although the difference is not substantial.

**Table 9:** The $\overline{\text{MMD}}$ for each network statistic on the BA dataset for samples generated by `DAMNETS` using different different numbers of GAT layers. Lower is better. We see that the performance is similar across GNN depths, with shallower networks performing slightly better.

| Number of Layers | Degree | Clustering | Spectral | Transitivity | Assortativity | Closeness |
|---|---|---|---|---|---|---|
| 1 | **$8e^{-3}$** | 0.78 | **0.14** | **0.01** | **0.01** | **$5e^{-6}$** |
| 2 | $9e^{-3}$ | **0.77** | 0.15 | 0.02 | **0.01** | $7e^{-6}$ |
| 4 | 0.04 | 0.91 | 0.22 | **0.01** | 0.06 | $4e^{-3}$ |

It is important to keep in mind that the GNN embeddings are just one component of the model pipeline, with the downstream task being generating samples at the next timestep. Therefore intuition that applies in other applications of GNNs such as deeper networks being better (up to the point of oversmoothing) may not apply here. In particular `DAMNETS` has no information other than the present state of the graph $G_t$. We hypothesise that if the GNN embeddings computed at time $G_t$ are similar to those computed at some other time $G_{t+k}$ (which may occur in a deeper network that oversmoothes the node features), then the model may become "confused" and sample the wrong type of transition, hence shallower networks perform better.

### B.4 Choice of Encoder

In Section 3.2 we mentioned that the self-attention layer in the decoder could be replaced with other recurrent modules. Here we study the performance of `DAMNETS` when using an LSTM instead of self-attention (which as before we call TFEncoder. Again we repeat the BA experiment, using a 3-layer LSTM or TFEncoder, the results of which are displayed in Table 10.

**Table 10:** The $\overline{\text{MMD}}$ for each network statistic on the BA dataset for samples generated by `DAMNETS` using different decoder layers. Lower is better. We used three layers for both the TFEncoder and LSTM. We see that the performance only degrades mildly when moving from the self attention layer through to the LSTM.

| Layer Type | Degree | Clustering | Spectral | Transitivity | Assortativity | Closeness |
|---|---|---|---|---|---|---|
| TFEncoder | **$8e^{-3}$** | **0.78** | **0.14** | **0.01** | **0.01** | **$5e^{-6}$** |
| LSTM | $9e^{-3}$ | 0.91 | 0.20 | 0.02 | 0.03 | $4e^{-5}$ |

We see a slight degradation in performance when moving from self-attention to LSTM. This is to be expected - the self attention layer can directly attend over all the tokens (here node-embeddings) in it's receptive field, whereas the LSTM only models dependencies via the hidden state. LSTM layers use significantly less memory however, so for large graphs with many nodes this may be advantageous. Another alternative is the Fenwick Tree structure introduced in [11].

# C   Experimental Details

## C.1   Model Specification and Training Details

For the experiments in this paper we used a hidden size of $F = 256$ for all experiments, as this was the default hidden size used in BiGG [11]. We used a single layer GAT [15] for all experiments. The rationale for this is as follows: GNNs are known to suffer from an *oversmoothing* problem [35], whereby node embeddings all become similar when using many stacked GNN layers. This would be particularly problematic in our case, as the model would not be able to distinguish between different states in the Markov chain and would likely perform very poorly. We therefore chose to use a very simple model with one GNN layer. It is possible this could be improved upon. All the LSTM networks in the BiGG decoder used 2-layers.

We used the Adam [36] optimiser for all experiments, with learning rate $0.001$ and weight decay parameter $0.0005$. We have not made an effort to optimise these parameters. We used early stopping based on the log-likelihood of the validation set, which was comprised of $30\%$ of the training data, chosen randomly. We used a batch size of 32 graphs (using gradient accumulation for the larger graphs to keep this consistent) and clipped gradients at a norm of $5$. We found that training to $0$ training loss was very harmful for out of sample performance, and that early stopping is necessary for good performance. All numerical results are averaged over five seeds.

We implemented the GAT using Torch Geometric [37], and used PyTorch [38] for the other deep learning functionality. We used Networkx [21] for processing the network data. We modified the original BiGG implementation to combine this with the encoder, which can be found at this link.

## C.2   Baseline Model Information

We used the publically released versions of DYMOND and TagGen. Both of these had fatal errors in their implementation, which we have fixed and released as a part of our source code. There is no available code for AGE, so we implemented this using standard PyTorch Transformer modules. We used all the default hyperparameters given in the respective papers. For training AGE we also used early stopping with the same validation log-likelihood criterion, batch size and optimiser settings. As AGE is a Transformer model, we experimented with many "tricks" that are commonly used to train Transformers, such as warmup learning rates as described in [7], but found they did not improve the performance of the model.

## C.3   Hardware and Running Time

All the experiments in this paper were carried out on a single Nvidia GeForce RTX 3090 GPU with an Intel(R) Xeon(R) Silver 4210R CPU @ 2.40GHz. For the smaller experiments we capped the run time of DAMNETS and AGE at one hour, and 24 hours for the larger datasets (although in all cases both these models early stopped before the cap). DYMOND is also fast to run despite needing to be re-trained on each NTS due to its simplicity. TagGen required 24 hours to complete its experimental run on all datasets.

# D Further Experimental Plots

Here we present further results for test statistics on the B-A dataset, the bipartite contraction model, the three-community decay model, and the test correlation networks.

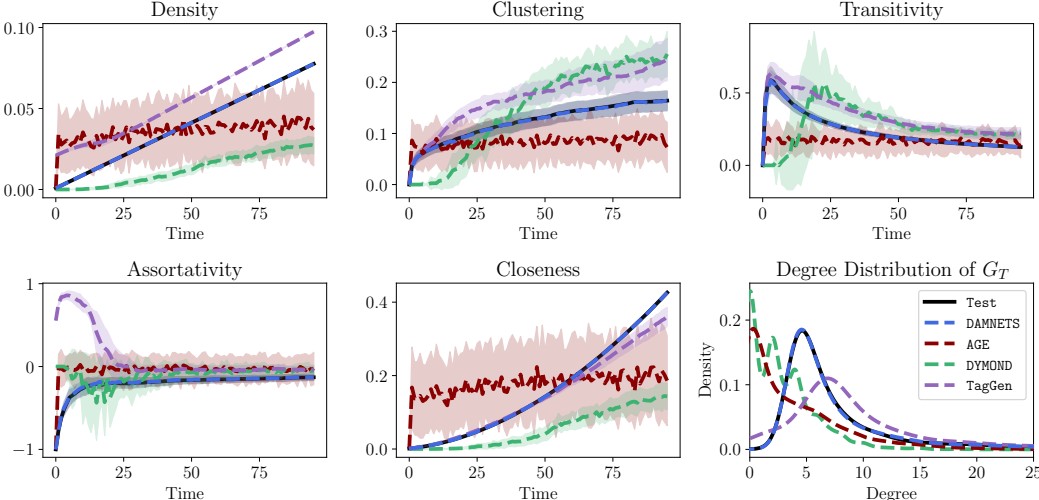

**Figure 8:** The first five plots show the mean and standard deviation of the network statistics computed through time for the B-A dataset. We see that DAMNETS produces samples that are very similar to the test set across all metrics, whereas the baseline methods fail to do so. The final plot shows the average degree distribution of the final network $G_T$ produced by the models. Only DAMNETS correctly replicates the power law degree distribution.

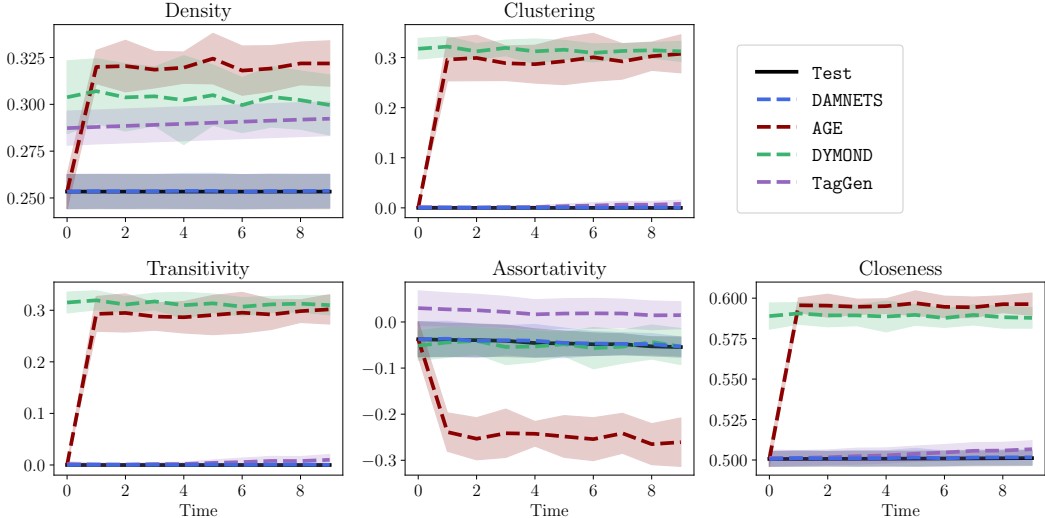

**Figure 9:** The network statistics computed through time for the bipartite contraction model. We see that DAMNETS shows excellect performance on all statistics, whereas the other models are not able to learn the dynamics.

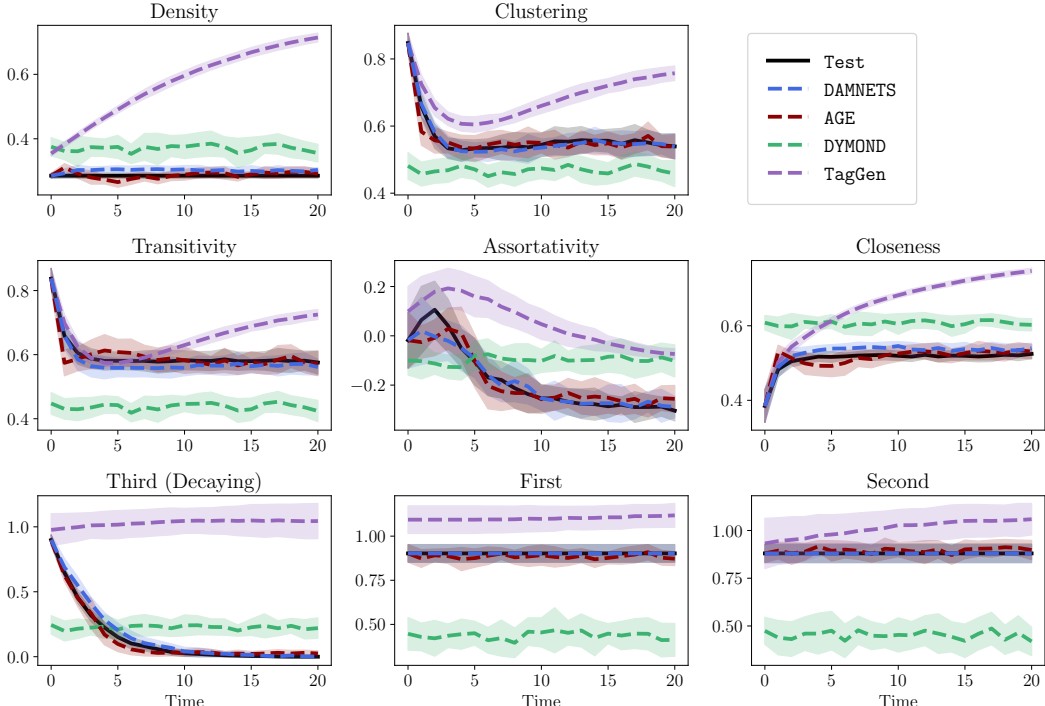

**Figure 10:** Statistics computed through time on the test set for the three-community decay model. First two rows: The average networks statistics computed across time. Final row: the density of each community through time. We see that both `AGE` and `DAMNETS` both show strong performance on this model, while `DYMOND` performs poorly.

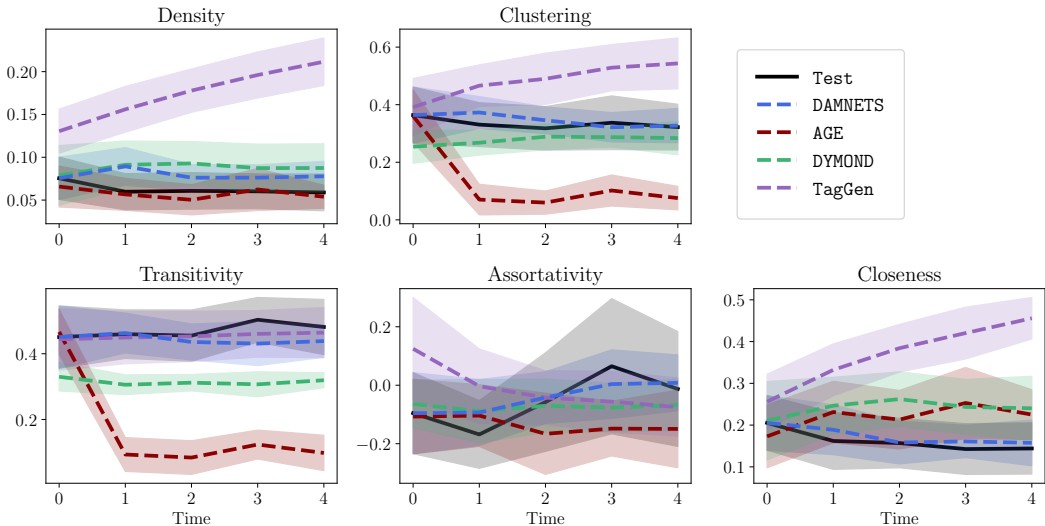

**Figure 11:** The average network statistics computed through time for the test correlation networks. We see `DAMNETS` closely tracks the test distribution on all statistics other than density.

