# OpenReview forum: "DAMNETS: A Deep Autoregressive Model for Generating Markovian Network Time Series"
_logconference.io/LOG/2022/Conference — LoG 2022 Poster_

### Official Review · Reviewer_VN3W · 2022-10-13

**Overall Score:** 6
**Confidence:** 4

**Review:**

**Contributions**

This paper introduces an architecture called DAMNETS for the forecasting of graph time series.

The main contribution of the paper is to frame the task as the prediction of a Markovian process $p(G_t | G_{t-1})$ and training a neural network to generate only the difference between subsequent adjacency matrices (ie, a delta matrix).

The architecture consists of a graph attention network to encode the current graph state and a modified version of the BiGG algorithm to sample the rows of the delta matrix as a function of the node encodings.

The model is then evaluated on three synthetic tasks with random graph models, and one real-world task with financial correlation networks.
The results show that the model outperforms three relevant baselines from recent literature, sometimes by a large margin.

**Strengths**

- The problem addressed in this paper is very interesting and is a major topic of research in the graph machine learning community.
- The method seems to provide significant advantages on the tested tasks.
- The design of the architecture is well-motivated and modular (e.g., the encoder could be improved/changed in the future to get better performance).
- The paper is well-written and easy to follow; the main ideas are explained clearly and the exposition is supported by figures and algorithms.

**Weaknesses**

- The paper could be contextualized better within the literature on graph time series prediction. To mention a few works that I believe fit well in Section 2:
	- Zambon et al., 2019, introduced several models for the auto-regressive prediction of graph stochastic processes.
	- Hajiramezanali et al., 2019, introduced a variational recurrent graph autoencoder for Markovian dynamic graphs.
	- Paassen et al., 2020, introduced a graph neural network model to predict graph edits between consequent time steps in a graph time series.

     In some sense, the main contributions of the paper are an instance of these previous ideas. Particularly, the Graph Edit Networks proposed by Paassen et al., 2020 are very relevant to the main contribution of DAMNETS of predicting the adjacency delta.
- DAMNETS are limited to predicting graphs where the number of nodes is fixed. They also do not support attribute prediction. These limitations should be discussed by the authors since they were already addressed in previous literature (so DAMNETS are making a step back in this aspect).
- The experimental results are convincing, but the method also performs sensibly worse on the real-world task. This raises an issue of how applicable these results are to other real-world problems. I advise the authors to expand their benchmark suite to include more real-world data (there are plenty of datasets available online -- e.g., see https://snap.stanford.edu/data/).

**Recommendation**

The paper is a nice contribution to the literature on graph time series and the results are convincing enough.

I am currently leaning towards a weak rejection, because of the weaknesses highlighted above.
However, I am willing to turn the recommendation into a weak (or even full) acceptance if the main issues above are addressed by the authors:

1. Address previous relevant work and compare it with the paper's contributions
2. Comment on the limitations of predicting fixed-size graphs and attributes and how these could be addressed (e.g., additional heads like Graph Edit Networks).
3. (If time allows) Add 2-5 experiments with real temporal networks to make results stronger.

**Questions**

- Using one-hot encodings as surrogate node features implies a fixed ordering of the nodes. This is likely to have an impact on the prediction if the nodes are permuted (eg, at test time we might assume that the node list changes order without us knowing the correspondence between the training and test node order). Have the authors considered this possibility? As a possible solution, Laplacian positional encodings could be used instead of node ids.
- The assumption that the time series are Markovian should be validated. The synthetic tasks are Markovian by design, so we only really have one data point to verify the assumption. Have the authors explored the literature to see if this question was addressed before?

**References**

Zambon, Daniele, et al. "Autoregressive models for sequences of graphs." 2019 International Joint Conference on Neural Networks (IJCNN). IEEE, 2019.

Hajiramezanali, Ehsan, et al. "Variational graph recurrent neural networks." Advances in neural information processing systems 32 (2019).

Paassen, Benjamin, et al. "Graph edit networks." International Conference on Learning Representations. 2020.

---

### Official Review · Reviewer_mucD · 2022-10-20

**Overall Score:** 8
**Confidence:** 3

**Review:**

This paper presents a generative model for learning how dynamic graphs evolve over time. This is an interesting and relatively little studied problem, but which has many potentially interesting applications. This model has two key advances over existing methods such as AGE.
1. The authors train a model to predict the difference between the current graph and the next graph in the time series. These delta graphs are often sparse and provide a simpler training objective.
2. Leveraging the sparsity of the delta graph, the authors develop a sparse generator which is more efficient than existing methods which generate an entire graph (nxn adjacency matrix) at once

Strengths:
1. The paper is clearly written and tackles a new and interesting problem.
2. The model is carefully considered and well-motivated with reference to the literature
3. The sparse decoder is, to my knowledge, a relatively novel construct
4. The authors show strong performance in comparison to existing methods on a number of synthetic benchmarks

Weaknesses:
1. All but one of the experiments are on synthetic data. While this is not a problem per se, especially given the novelty of the problem, the paper would be strengthened by evaluation on real data. For example, it looks like the AGE paper considered a number of real-world "graph evolution in time" data-sets. Why were the authors not able to make use of these? **This has been addressed in author rebuttal so I raise my score to 8**

---

### Official Review · Reviewer_AECm · 2022-10-21

**Overall Score:** 6
**Confidence:** 3

**Review:**

## Summary:

This paper introduces DAMNETS, a scalable deep generative model for network time series (NTS). DAMNETS maintains scalability via generating delta matrices instead of entire graphs as prior works did. The delta matrices are usually sparser than the full adjacency matrix, enabling the model to utilize sparse sampling methods. Different from prior methods for NTS generation, DAMNETS is able to explicitly utilize graph connectivity in the model pipeline and model edge correlations within rows of the adjacency matrix. Overall, it's a good paper.


## Reasons for score

The problem this work targets is of practical significance to the LoG community. The proposed method is well-motivated. The presentation of each section is quite clear and well-organized. Therefore, it's easy for readers to understand the novel idea and elegant design of the new framework. Sufficient and extensive experiments over different datasets and ablation study well demonstrate the effecitiveness. In addition, the experimental details are clearly presented. Overall, I vote for accepting.

## Pros:

- This paper proposes a solution to a significant problem, i.e. NTS generation, which has many application scenarios in reality. For example, it helps simulating the trajectories of future pandemics , especially for the covid-19.
- The motivation is reasonable and the proposed method is sufficiently novel. It samples each row using a binary tree, bringing about significant time savings. Besides, it proposes to generate delta metrices rather than entire graphs. Hence, the good scaleability is maintained.
- Extensive experiments show that DAMNETS yeilds consistent and significant improvement on several datasets. Ablation studies further verify the effectiveness of the proposed method. The details of experiments are quite clear.



## Cons:

- DAMNETS adopts a standard Transformer self-attention layer. However, as mentioned in Sec. 3.2, self-attention does not scale well to very long sequences. The authors propose to adopt an LSTM or the Fenwick Tree instead for large graphs comprising of many nodes. It could be interesting to verify this idea on larger graphs.
- As described in Appendix C.1, only one-layer GAT is adopted. Although GNNS are known to suffer from an oversmoothing problem when stacking many GNN layers, I think using one layer might be inadequate for capturing informative neighbored information. It would be more convincing to see the results of using more layers and investigate the sensitivity of the proposed method to the layer number. What's more, I believe the contribution can be further improved if there is more experiments w.r.t. more GNNs choices to verify the sensitivity to GNNs architectures.


## Minor concern:

I suggest summarizing and highlighting the contributions of this paper in the end of introduction part, i.e. Sec. 1.

---

### Meta-Review · Area_Chair_498b · 2022-11-21

**Confidence:** 5
**Recommendation:** Accept

**Meta Review:**

This manuscript introduces a scalable generative approach for time series of networks, an important problem with many applications that have received relatively little attention in the literature. Broad discussion between PC members and the authors showed numerous strengths of this paper that are key drivers of my recommendation:
* S1: Motivation is clear. It tackles a new and interesting problem, the new method is sufficiently novel. The problem addressed in this paper is very interesting and is a major research topic in the graph ML field.
* S2: The paper is clearly written. The main ideas are explained clearly and supported by figures and algorithms.
* S3: Experiments comprehensively compare the new approach to relevant baselines on synthetic benchmarks.

The authors carefully considered questions raised by the reviewers and addressed the points by adding a new dataset to the experimental section, new ablation studies, and expanded the related work section with a discussion on network time series forecasting.

---

### Decision · Program_Chairs · 2022-11-22

Accept (Poster)